# Integrated Metabolomic and Transcriptomic Analyses Reveals Sugar Transport and Starch Accumulation in Two Specific Germplasms of *Manihot esculenta* Crantz

**DOI:** 10.3390/ijms24087236

**Published:** 2023-04-13

**Authors:** Jie Cai, Jingjing Xue, Wenli Zhu, Xiuqin Luo, Xiaohua Lu, Maofu Xue, Zhuowen Wei, Yuqi Cai, Wenjun Ou, Kaimian Li, Feifei An, Songbi Chen

**Affiliations:** 1Tropical Crops Genetic Resources Institute, Chinese Academy of Tropical Agricultural Sciences, Key Laboratory of Ministry of Agriculture for Germplasm Resources Conservation and Utilization of Cassava, Haikou 571101, China; caijie@catas.cn (J.C.); xuetao608@163.com (J.X.); zhuwenbamboo@126.com (W.Z.); xiuqinluo@163.com (X.L.); xuemaofu@163.com (M.X.); zhuowen936@163.com (Z.W.); cassava6973@163.com (W.O.); likaimian@sohu.com (K.L.); 2School of Life Science, Hainan University, Haikou 570228, China; 19071010110006@hainanu.edu.cn (X.L.); yuqicai87@163.com (Y.C.)

**Keywords:** cassava, SWEETs, sugar transport, starch accumulation, metabolomics, transcriptomics

## Abstract

As a starchy and edible tropical plant, cassava (*Manihot esculenta* Crantz) has been widely used as an industrial raw material and a dietary source. However, the metabolomic and genetic differences in specific germplasms of cassava storage root were unclear. In this study, two specific germplasms, *M. esculenta* Crantz cv. sugar cassava GPMS0991L and *M. esculenta* Crantz cv. pink cassava BRA117315, were used as research materials. Results showed that sugar cassava GPMS0991L was rich in glucose and fructose, whereas pink cassava BRA117315 was rich in starch and sucrose. Metabolomic and transcriptomic analysis indicated that sucrose and starch metabolism had significantly changing metabolites enrichment and the highest degree of differential expression genes, respectively. Sugar transport in storage roots may contribute to the activities of sugar, which will eventually be exported to transporters (SWEETs), such as (MeSWEET1a, MeSWEET2b, MeSWEET4, MeSWEET5, MeSWEET10b, and MeSWEET17c), which transport hexose to plant cells. The expression level of genes involved in starch biosynthesis and metabolism were altered, which may result in starch accumulation. These results provide a theoretical basis for sugar transport and starch accumulation and may be useful in improving the quality of tuberous crops and increasing yield.

## 1. Introduction

Cassava (*Manihot esculenta* Crantz) is one of the most important staple crops in developing countries because of its starchy roots that provide essential dietary carbohydrates [1,2,3,4]. In recent decades, some specific cassava germplasms from Brazil have been introduced and preserved in the National Cassava Germplasm Repository in China. For instance, specific germplasms of sugar cassava with white flesh (SC), pink cassava with pink flesh (PC), and yellow cassava with yellow flesh (YC), are related to storage roots originating from Brazil (EMBRAPA) [5]. The characteristics of cassava germplasms varied by types and contents of chemical constituents, especially carbohydrates and starch [6,7]. However, metabolomic and genetic differences in sugar transport and starch accumulation of cassava storage roots in these specific germplasms are poorly understood.

Sink capacity depends on the delivery of energy sources (carbohydrates) by the phloem in cassava [8]. As the main photo-assimilate, sucrose is transiently stored in mature leaves, and then it is delivered to organs by the active phloem [9]. Notably, SWEETs (“sugar will eventually be exported transporters”) as regulators of sugar fluxes, transporting sugar across membranes during source-to-sink processes. SWEET can be distinguished according to the types of sugar it transports—clade I and II members, which transport hexose (priority to transport glucose), clade III members, which transport sucrose across membranes, and clade IV members, which transport fructose [10]. SWEETs are not only involved in sugar transport in plant cells, but also play vital roles in plant growth and development [11]. They facilitate sucrose and starch accumulation in *Arabidopsis thaliana* and *Pyrus ussuriensis* [12,13], transport hexose by basal endosperm transfer layer in *Zea mays* [14], transport glucose in *Medicago truncatula* and *Oryza sativa* [15], sucrose influx and efflux in *Glycine max* [16], and sucrose leakage in *Solanum tuberosum* [17]. Research on SWEETs of cassava is rarely reported. Recent studies have shown that the gene cloning and subcellular localization of SWEETs [18,19,20] are involved in various types of stress and pathogen interaction in cassava [21,22,23], but the critical roles of SWEETs in source and storage tissues remain poorly understood in cassava.

To better understand the biological process of sugar transport and starch accumulation in the specific germplasms, we investigated the transcript and metabolic profiles of SC, which were rich in sugar, and PC, which were rich in starch. The variations in starch and sucrose metabolism, and other metabolites, including the precursors of starch, were investigated through metabolomic and transcriptomic analyses in this study.

## 2. Results

### 2.1. Measurement of Total Soluble Hexose and Starch in Different Cassava Storage Roots

The phenotypes of different cassava storage roots were presented (Figure 1). The contents of total starch, sucrose, glucose, and fructose were determined in the storage roots of two specific cassava germplasms, which were measured (Table 1). The results showed that the content of total starch of SC and PC were 11.03% and 29.04%, respectively. The content of sucrose in PC was 12,850.65 mg/kg, which was significantly lower than that of SC (16,049 mg/kg, *p* ≤ 0.05). PC had significantly higher glucose content (8725.70 mg/kg) than SC (1707.52 mg/kg), but lower fructose content (1756.78 mg/kg) than SC (9874.10 mg/kg, *p* ≤ 0.05).

### 2.2. Metabolomic Analysis of LC-MS Data of All Storage Roots of Cassava

In a LC-MS untargeted metabolomics analysis, a total of 2705 metabolites were detected and quantified in the different storage roots of cassava. Principal component analysis (PCA) and cluster analysis were used in elucidating the overall metabolic differences and metabolite accumulation patterns between SC and PC. According to the PCA score map, the metabolites of SC and PC could be clearly distinguished (Appendix A).

As shown in the volcano map, a total of 299 SCMs were detected in SC and PC (124 up-regulated and 175 down-regulated) (Figure 2A; Appendix A). These SCMs were generally classified into 12 categories, including primary metabolites, such as lipids, lipid-like molecules, organic acids, organic derivatives, organic oxygen compounds, phenylporpanoids, and polyketides (Figure 2B). A heatmap cluster analysis was also performed for the characterization of the SCMs (Figure 2C, Top 50). Notably, 23 of the 26 SCMs of organic oxygen compounds were carbohydrates. Using the identification criterion of *p* ≤ 0.05 and VIP value ≥ 1, the content of 23 carbohydrates (six up-regulated and seventeen down-regulated) significantly varied between PC and SC.

### 2.3. Differential Expression of Genes between SC and PC

After strict quality control and data screening of transcriptome sequencing, 38.53 G clear data were obtained. The Q30 value was over 91.42%, the average content of GC was 43.02%, and the sequencing error rates were lower than 0.1%. These results indicated that the quality of the sequencing data was good enough for further analysis (Appendix A).

An amount of 23,268 genes were detected by transcriptome sequencing. Through screening and filtering, a total of 9077 DEGs were obtained between SC and PC, including 4779 DEGs that were up-regulated and 4298 DEGs that were down-regulated, respectively. The volcano plots and fold changes were used in visualizing the overall distribution of gene expression levels (Figure 3A and Appendix A).

GO and KEGG enrichment analysis indicated that DEGs were significantly associated with biological process, cellular components, and molecular functions in SC and PC (Figure 3B). In the KEGG annotation database, some DEGs were mainly enriched in starch and sucrose metabolism, galactose metabolism, glycolysis/gluconeogenesis, etc. (Figure 3C).

### 2.4. Correlation Analysis of the Metabolome and Transcriptome

Correlation analysis on the SCMs and DEGs were calculated. A total of 8176 correlation pairs had |Correlation| ≥ 0.8 (Appendix A). Then, the correlation calculation results of SCMs and DEGs were plotted in a clustering heatmap (Top 20, Appendix A). Based on the correlation analysis, network analysis of SCMs and DEGs were found and illustrated in PC compared to SC (Appendix A). Especially, in sucrose and starch metabolism, the SCMs, such as sucrose and D-glucose-6-phosphate, were up or down regulated by the DEGs (Figure 4 and Appendix A). The above SCMs and DEGs may play an important role in the differences in sugar transport between SC and PC in cassava.

### 2.5. Alterations of Sugar Transport by SWEETs between SC and PC

The expression profile of six SWEETs were found in PC compared to SC, four SWEETs (*MeSWEET1a*, *MeSWEET2b*, *MeSWEET4* and *MeSWEET5*) were up-regulated, and two SWEETs (*MeSWEET10b* and *MeSWEET17c*) were down-regulated. The qRT-PCR results confirmed the transcriptomic data and showed that the expression levels of *MeSWEET1a* and *MeSWEET5* were significantly higher in PC (Figure 5).

### 2.6. Sugar Transport Activity by Five SWEETs between SC and PC

Five SWEET genes (*MeSWEET1a*, *MeSWEET2b*, *MeSWEET4*, *MeSWEET10b,* and *MeSWEET17c*) were expressed in the yeast mutant EBY.VW4000. Yeast cells expressing either MeSWEET1a or MeSWEET4 could grow on media supplemented with glucose, fructose and sucrose, suggesting that MeSWEET1a and MeSWEET4 can transport these three types of sugars in yeast, but both demonstrated high transport activity for glucose. MeSWEET2b showed weak activity for glucose transport. The expression of MeSWEET10b effectively restored the growth of EBY.VW4000 on media supplemented with sucrose. MeSWEET17c can transport sucrose, glucose, and fructose in yeast, especially with high transport activity for fructose (Figure 6).

### 2.7. Alterations in Starch Biosynthesis in Storage Roots of SC and PC

In this study, starch content significantly increased in PC (Table 1). The expression levels of 11 genes related to starch metabolism were found in PC compared with SC, two genes (*MeAGPase* and *MeSPS1*) were up-regulated, and nine genes (*MeAGPL*, *MeGBSSI*, *MeSSII-1*, *MeSP*, *MeSP1*, *MeSP2*, *MeSPS3*, *MeISA2,* and *MeAMY3-2*) were down-regulated. The qRT-PCR results confirmed the transcriptomic data and indicated that the expression level of starch biosynthetic gene *MeSPS1* significantly increased, meanwhile, the expression levels of starch metabolic genes significantly decreased (Figure 7). These results may be contributed to high starch accumulation in PC.

## 3. Discussion

Metabolomics is an effective method for determining the metabolites in plants and analyzing metabolites in different tissues of cassava [24,25]. The composition of metabolites can directly affect the nutritional quality of storage roots in cassava. In this study, 299 SCMs were identified by the non-targeted metabolome in SC and PC. The quantity of SCMs in this study was significantly different from that in previous research into storage roots [26,27], owing to the characteristics of different cassava varieties. The composition and content of metabolites in cassava storage roots can directly affect the quality. In this study, the essential amino acids L-arginine, L-histidine, L-tyrosine, and tryptophan accumulated in PC compared with SC, which are indispensable to human health. In addition, the content of scopoletin was significantly higher in PC. This compound is involved in postharvest physiological deterioration (PPD) in cassava. This result indicated that PC had strong ability to resist PPD compared with SC, which was in accord with previous studies [28,29,30]. In the future, scopoletin will be used as a biomarker for PPD, and PC will be used as a material for cassava breeding. The accumulation of metabolites in plants depends on plant genetic factors. Understanding these important factors and their functions will help in increasing yield and improving quality in tuberous crops. In this study, two specific germplasms (SC and PC) were used as research materials for the investigation of the difference between metabolite accumulation and gene expression. Combined analysis of metabolomic and transcriptomic data in this study indicated that DEGs and SCMs were enriched in sucrose and starch metabolism pathways (Figure 3). Sucrose as the primary sugar is delivered by phloem by unloading and loading activities between source and sink tissues [31,32,33]. The efflux of sucrose was transported by the SWEET transporter family, which is involved in sources and sink strength [33]. In previous reports, many studies have shown that starch accumulation in cassava storage roots is associated with the high expression levels of sugar transporter genes in the stems. Therefore, increasing the expression levels of sugar transporter genes in sink tissues can enhance phloem-loading capacity, can solve the negative feedback loop of the photo-assimilates, and can increase crop yield [2].

Cassava is a source-to-sink tuberous crop, and the source from photosynthesis in the leaves directly affected the sink capacity of cassava storage roots [34,35]. SWEETs play an important role in plant growth and development. An amount of 28 SWEETs were analyzed in cassava, and then phylogenetic analysis indicated that most SWEETs (12 SWEETs) belong to the clade III member [19]. In this study, sucrose content was significantly reduced. According to the transcriptomic data, *MeSWEET10b*, which is responsible for transporting sucrose, belongs to the clade III member, which was down-regulated (Figure 5) in PC compared with SC (Figure 6). These results were consistent with previous work and suggested that the clade III member of SWEETs are involved in cassava storage root development [36]. Simultaneously, *MeSWEET17c*, which belongs to the clade IV member and is responsible for transporting fructose, was down-regulated in PC compared with SC (Figure 5). Down-regulation of *MeSWEET10b* and *MeSWEET17c* may result in low sucrose and fructose accumulation in the storage root of PC (Table 1). Furthermore, the expression levels of *MeSWEET1a*, *MeSWEET2b,* and *MeSWEET5* (10-40 folds increase) were significantly up-regulated in PC compared with SC (Figure 5), and *MeSWEET4* was up-regulated in PC with a three-fold increase. These SWEETs preferentially transport glucose (Figure 6), and the high expression levels of *MeSWEET1a*, *MeSWEET4,* and *MeSWEET5* may promote glucose accumulation in the storage roots of PC (Table 1). In conclusion, sugar transport in cassava storage roots of different cassava germplasms depends on the activities of different SWEETs.

Starch accumulation in cassava storage roots is an extremely complex process, and it not only has a correlation with high expression levels of SWEETs, but it is also involved in starch biosynthetic enzymes. As a result, the content of starch in PC was higher than in that in SC (Table 1), and the metabolite glucose-6-phosphate was down-regulated in PC, which is precursor to starch (Appendix A). In addition, the expression level of starch biosynthetic gene *MeSPS1* was significantly up-regulated in PC. In conclusion, these results may be attributed to promoting the starch biosynthesis in PC (Figure 7).

SWEETs not only are involved in sugar transport in plants, but also play vital roles in seed filling, fruit development, biotic stress, abiotic stress, and symbiosis [37,38,39,40,41]. In general, cassava, with its starchy root, served as staple food in terms of calorie supply for raising nearly a billion people in the tropics and the subtropics, but it lacks nutrient elements [42]. Interestingly, if the function of a SWEET in cassava related to β-carotenoid accumulation, when the overexpression of this *SWEET* will be enhanced β-carotenoid accumulation, and contributed to the alleviation of vitamin A deficiency in tuberous crops.

## 4. Materials and Methods

### 4.1. Plant Materials

In this study, sugar cassava (SC) GPMS0991L and pink cassava (PC) BRA117315 were selected as materials. *M*. *esculenta* Crantz cv. Sugar Cassava GPMS0991L (SC) with green leaves and white flesh of storage roots and *M*. *esculenta* Crantz cv. Pink Cassava (PC) BRA117315 with green leaves and pink fleshy storage roots were used in this study, which were provided by the National Cassava Germplasm Repository of Tropical Crops Genetic Resources Institute (TCGRI), as well as the Chinese Academy of Tropical Agricultural Sciences (CATAS) in China. These plants were grown for 10 months in the field (Danzhou City, Hainan Province, China; 19°30′33.13″ N, 109°30′19.34″ E). Fresh plant material (fleshes of cassava storage roots; 10 months) from harvested plants were collected into 50 mL centrifuge tube in January 2020, and then all materials were frozen in liquid nitrogen and stored at −80 °C for RNA and metabolite extraction. All experiments were performed in three biological replicates in this study.

### 4.2. Measurement of Soluble Sugar Content

Ethanol (80%) was added to 0.2 g fresh tuberous roots. After centrifugation, the supernatant was collected and then dried in 1.5 mL centrifuge tube. The derivatization conditions were as follows: 30 μL of 20 mg mL^−1^ methoxyamino pyridine hydrochloride was added to dried samples, the mixture was shaken for 1.0 h at 37 °C and 650 rpm min^−1^, and 70 μL of N, O-bis-trifluoroacetamide was added. The mixture was shaken at 70 °C for 1 h at 650 rpm min^−1^, and it was kept at room temperature for 30 min. The supernatant was used for determination. A GC-MS column (Agilent, Santa Clara, CA, USA) and capillary column (HP-5, 30 mm × 0.25 mm, 0.25 μm) were used in the GC-MS analysis, which was performed at gasification chamber temperature of 280 °C. The temperature procedure was as follows: 80 °C for 1 min, from 6 °C/min to 240 °C, and maintained for 15 min. The carrier gas was helium, and the flow rate was 1 mL/min. The split ratio was 20:1, and the injection volume was 1 μL. Gas chromatography was connected to a mass spectrometry 5975 C quadrupole mass spectrometer.

### 4.3. Measurement of the Total Starch Content

The starch content was measured with a starch assay kit (BC0705, Solarbio, Beijing, China) according to the manufacture’s protocol.

### 4.4. Metabolite Extraction Procedure

Freeze-dried flesh was crushed with small two steel balls for 1 min at a frequency of 60 Hz. Then, 100 mg powder was weighed and extracted with 70% methanol aqueous solution. The aqueous solution was centrifuged at 10,000× *g* for 15 min, and the extracts were dried in a freeze concentration centrifugal dryer, and then they were dissolved in 20% methanol aqueous solution, and then they were filtered through a 0.22 μm microfiltraction membrane before LC-MS analysis.

### 4.5. Detection and Analysis of Metabolites

The ACQUITY UPLC I-Class system (Waters, Framingham, MA, USA) was coupled with the VION IMS QTOF mass spectrometer (Waters, Milford, MA, USA), which was used in analyzing the metabolic profile of positive and negative modes. UPLC analysis was performed using a UPLC column, Waters (Shanghai, China) ACQUITY UPLC BEH C18 column (1.7 μm, 2.1 × 100 mm). The solvent system was water (0.1% formic acid, solvent A) and acetonitrile/methanol (0.1% formic acid, solvent B). The gradient program was set as follows: 0 min, 1% B, 1 min, 30% B, 2.5 min, 60% B, 6.5 min, 90% B, 8.5 min, 100% B, 10.7 min, 100% B, 10.8 min, 1% B and 13 min, 1% B, flow rate, and 0.40 mL min^−1^; injection volume was 5 μL.

AB SCIEX Triple TOF 5600 System (AB SCIEX, Framingham, MA, USA) was used to analyze the metabolic profiles of electron spray ionization (ESI) positive and negative ion modes. In the positive ion mode, the capillary and sampling cone voltages were 2 kV and 40 V, respectively, compared with 1 kV and 40 V in the negative ion mode. Mass spectrometry data were collected in centroid MSE mode. The TOF mass range was set from 50 Da to 1000 Da, and the scan time was 0.2 s. For MS/MS detection, all precursors were fragmented using 20–45 eV and scanned for 0.2 s. During the acquisition, the LE signal was acquired every 30 s for the calibration of the mass accuracy. Furthermore, to evaluate the stability of the LC-MS system during the entire acquisition process, a quality control (QC) sample was acquired after every 10 samples were collected.

The acquired LC-MS raw data were analyzed by the Progenesis QI software v2.3 (Waters, Milford, CT, USA). The following parameters were used: precursor tolerance, 5 ppm, fragment tolerance, 10 ppm, and retention time (RT) tolerance, 0.02 min. Internal standard detection parameters were deselected for peak RT alignment, isotopic peaks were excluded for analysis, noise elimination level was 10.00, and the minimum intensity was 15% of base peak intensity. The Excel file containing the dimension data sets including *m*/*z*, peak RT, peak intensities, and RT–*m*/*z* pairs were used as the identifier for each ion. The resulting matrix was further reduced by removing any peaks with a missing value (ion intensity = 0) in more than 50% samples. The internal standard was used for data QC.

Metabolites were identified by the Progenesis QI (Waters, Milford, CT, USA) data processing software. Public databases, such as human metabolome database (HMDB) and self-built databases, were used. Positive and negative data were combined and imported into a R package. Principle component analysis (PCA) and partial least-squares-discriminant analysis (PLS-DA) were carried out for the visualization of metabolic alterations among experimental groups, after mean centering and Pareto variance scaling, respectively. Variable importance in the projection (VIP) ranks the overall contribution of each variable to the orthogonal partial least-squares-discriminant analysis (OPLS-DA) model, and variables with VIP > 1 and *p* < 0.05 were considered relevant for group discrimination.

### 4.6. RNA Extraction and Illumina Sequencing

Total RNA was extracted from frozen tuberous roots with a RNAprep Pure Plant Kit (Tiangen Biotech, Beijing, China). RNA quality was determined by running an agarose gel with GelStain (TransGen, Beijing, China) staining. RNA concentration was determined with NanoVueTM Plus ultramicro spectrophotometer (GE Healthcare, New York, NY, USA). Poly (A) mRNA was enriched from total RNA using Oligo (dT) magnetic beads. Poly (2×) RNA was subsequently fragmented and then transcribed into first-strand cDNA using reverse transcriptase and random hexamer primers. Second-strand cDNA was synthesized using second strand marking master mix (Invitrogen, Carlsbad, CA, USA). After end repair and the addition of a poly (A) tail, fragments with suitable length were isolated and connected to sequencing adaptors. The fragments were sequenced on an Illumina HiseqTM 2500 platform.

### 4.7. Transcriptome Data Analysis and Annotation

To acquire high-quality reads, the raw reads of fastq format were processed using Trimmomatic, and low-quality reads were removed. Clean reads for each sample were retained for subsequent analysis. Gene function was annotated by the Kyoto Encyclopedia of Genes and Genomes, Gene Ontology, Swiss-Prot, trEMBL, and KOG databases. The FPKM of each gene was calculated as an indicator for measuring transcript or gene expression levels. Genes with a |log2fold change| ≥ 1 and FDR < 0.05 were identified as differentially expressed genes (DEGs) by using DESeq2. GO and KEGG enrichment analysis of DEGs were performed, respectively, using R, based on the hyper genometric distribution.

### 4.8. Correlation Analysis between Transcriptome and Metabolome Data

Pearson correlation coefficients were calculated to integrate transcriptome and metabolome data. In this study, for the joint analysis between the transcriptome and metabolome, the screening criterion was a Pearson correlation coefficient great than 0.8.

### 4.9. qRT-PCR Analysis of Genes Involved in Sugar Transport and Starch Metabolism

Total RNA was extracted from 200 mg of tuberous roots using a RNAprep Pure Plant Plus Kit (Tiangen, Beijing, China). One-Step gDNA Removal and cDNA Synthesis SuperMix (TransGen, Beijing, China) were used for genomic DNA digestion and first-strand cDNA synthesis. Reverse transcription was performed according to the manufacturer’s protocol. Specific primers for genes involved in carbohydrate biosynthesis and *MeActin* gene (internal control) are listed in Appendix A. qRT-PCR reactions were performed in 10 µL volume in a Piko REAL thermocycler (Thermo Fisher Scientific Inc., Göteborg, Sweden). For each gene, all experiments were performed in triplicate per sample. The comparative CT method (2^−ΔΔCt^) was used in quantifying gene expressions.

### 4.10. Complementation of Yeast EBY.VW4000

The ORF of *MeSWEETs* was amplified (specific primers are listed in Appendix A) from pEASY-Blunt-*MeSWEETs* clones and then cloned into pDR196 vector. The pDR196-*MeSWEETs* construct was then used to transform in *Saccharomyces cerevisiae* EBY.VW4000, which is completely deficient in hexose uptake due to multiple mutations in hexose transporters, but it can grow on maltose medium [43]. The EBY.VW4000 strain was grown on 1% yeast extract/2% peptone medium supplemented with 2% maltose. After transformation, the yeast cells were streaked on SD-Ura medium and 2% maltose. For complementation growth assays, cells were grown overnight in liquid SD (-Ura) medium supplemented with 2% maltose. Serial dilutions (OD_600_ = 0.1, 0.01, 0.001) were plated on SD (-Ura) medium, containing either 2% maltose (as the control) or 2% sucrose, 2% glucose, or 2% fructose. The plates were incubated at 30 °C for two to five days and photographed.

### 4.11. Statistical Analysis

The original data were compiled using MS Excel 2010 software (Microsoft, Redmond, WA, USA), and R software (version 3.6.0) under the terms of the Free Software Foundation’s GNU General Public License in source code form was used for data analyses. All data were expressed as the mean ± standard error (SE). All data were from three biological replications.

## 5. Conclusions

Comparative metabolomics and transcriptome analyses of two specific germplasms were performed, and metabolites and genes in sugar transport and starch accumulation were identified. The expression levels of sugar transport proteins SWEETs and starch biosynthetic genes were significantly changed in SC and PC, and SWEETs genes were highly expressed on PC, which means a much larger amount of carbohydrate moves to the roots in PC and results in starch accumulation. This study may facilitate future research on the regulatory mechanisms of sugar transport and starch accumulation in cassava.

## Figures and Tables

**Figure 1 ijms-24-07236-f001:**
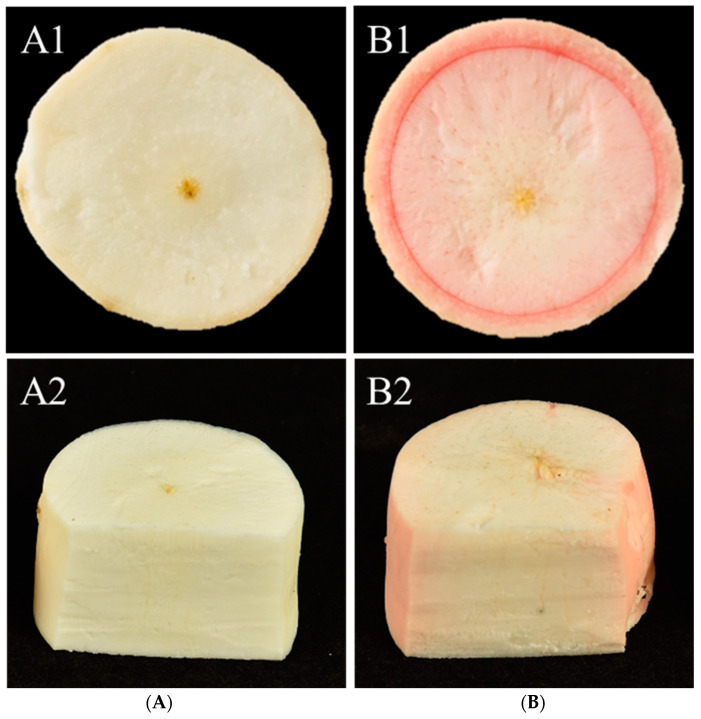
Phenotypes of SC and PC. (**A**) The white flesh of cassava storage roots; (**B**) The pink flesh of cassava storage roots. SC: *M. esculenta* Crantz cv. sugar cassava GPMS0991L; PC: *M. esculenta* Crantz cv. pink cassava BRA117315. (**A1**,**B1**) shows the of transverse section SC and PC, (**A2**,**B2**) shows the longitudinal section of SC and PC.

**Figure 2 ijms-24-07236-f002:**
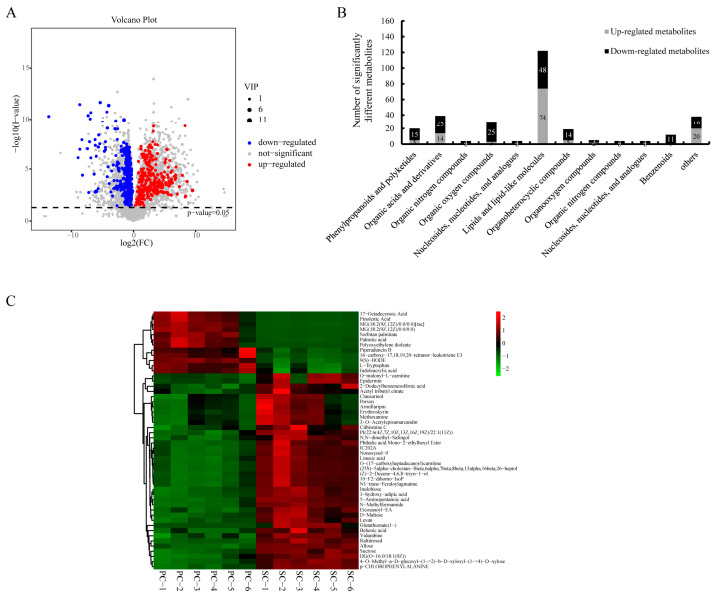
Significantly changed metabolites (SCMs) in PC compared to those in SC. (**A**) Volcano plot of the metabolites between SC and PC. (**B**) The number of SCMs in each category. (**C**) Heatmap of 299 SCMs. SC: *M. esculenta* Crantz cv. sugar cassava GPMS0991L; PC: *M. esculenta* Crantz cv. pink cassava BRA117315.

**Figure 3 ijms-24-07236-f003:**
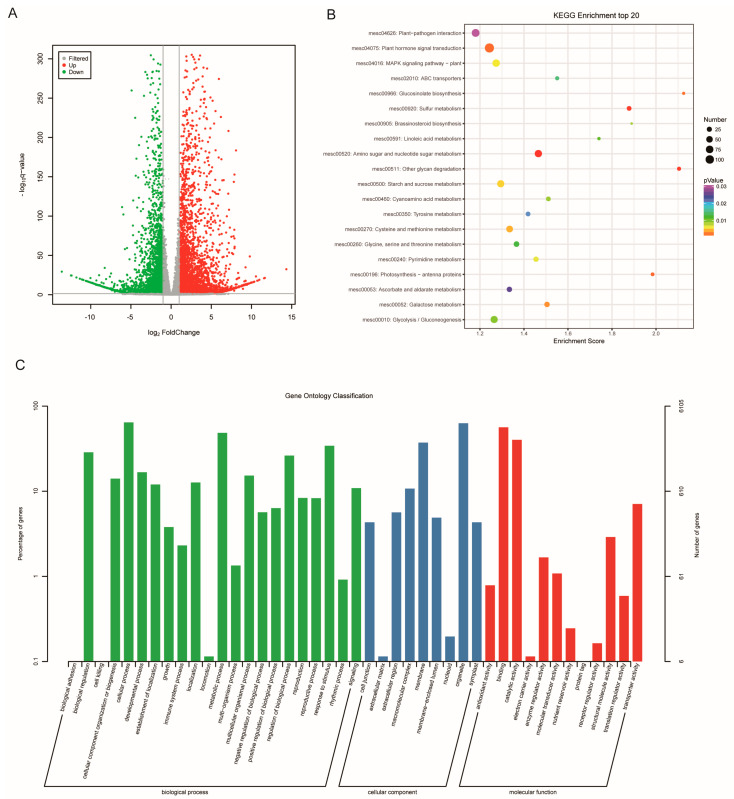
Screening and functional enrichment analysis of DEGs. (**A**) The overall distribution of gene expression levels and fold differences between SC and PC. (**B**) KEGG pathway enrichment of DEGs. (**C**) Gene ontology classification of DEGs. SC: *M. esculenta* Crantz cv. sugar cassava GPMS0991L; PC: *M. esculenta* Crantz cv. pink cassava BRA117315.

**Figure 4 ijms-24-07236-f004:**
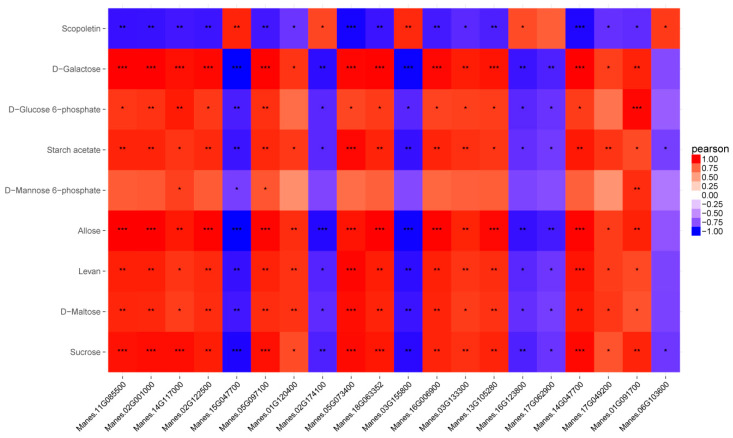
Correlation heatmap of SCMs and DEGs in sucrose and starch metabolism between SC and PC in cassava. Colored in “red” and “blue” indicated positive and negative correlations, respectively. SC: *M. esculenta* Crantz cv. sugar cassava GPMS0991L; PC: *M. esculenta* Crantz cv. pink cassava BRA117315. Correlation differences are indicated by asterisk (* *p* < 0.5, ** *p* < 0.01, *** *p* < 0.001).

**Figure 5 ijms-24-07236-f005:**
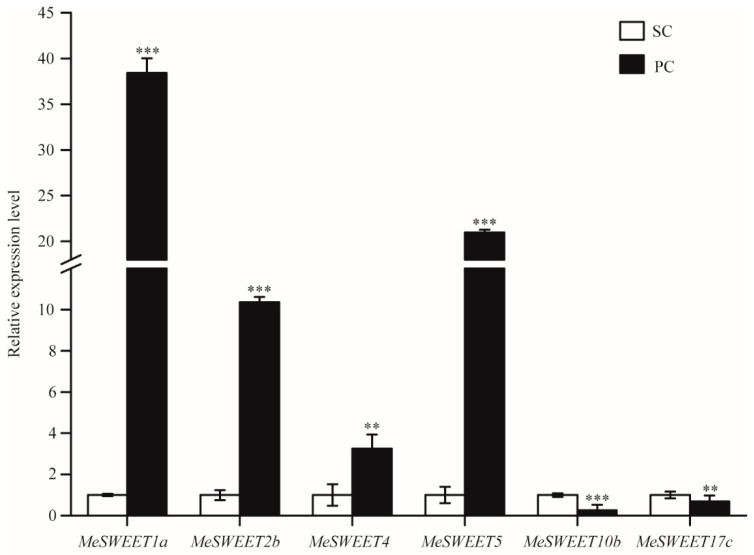
qRT-PCR analysis of the expression levels of *MeSWEETs* in PC compared to SC. The relative expression levels are normalized to *MeActin*. The x-axis represents the name of six *MeSWEETs* genes. The y-axis represents the relative expression levels of *MeSWEETs*. The data represent the means of three biological replicates. All data are the means ± SE of three independent experiments. Significant differences are indicated by asterisk (** *p* < 0.01, *** *p* < 0.001). SC: *M. esculenta* Crantz cv. sugar cassava GPMS0991L; PC: *M. esculenta* Crantz cv. pink cassava BRA117315.

**Figure 6 ijms-24-07236-f006:**
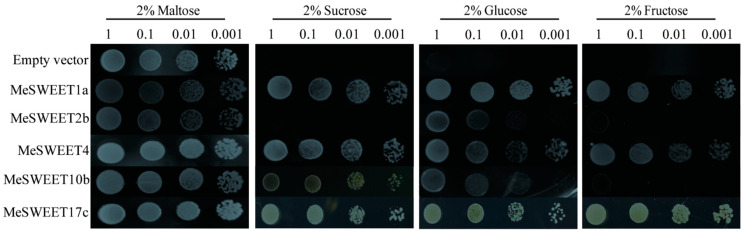
Transport activity of five MeSWEETs in the yeast mutant EBY.VW4000. Yeast cells expressing an empty vector or a vector containing *MeSWEET1a*, *MeSWEET2b*, *MeSWEET4*, *MeSWEET10b*, *MeSWEET17c* were diluted (10-fold) and cultured on selective synthetic deficient media without uracil (SD-Ura) supplemented with 2% (*w*/*v*) maltose, 2% (*w*/*v*) sucrose, 2% (*w*/*v*) glucose, or 2% (*w*/*v*) fructose as the sole carbon source. Images were captured after incubation at 30 °C for two to five days. SC: *M. esculenta* Crantz cv. sugar cassava GPMS0991L; PC: *M. esculenta* Crantz cv. pink cassava BRA117315.

**Figure 7 ijms-24-07236-f007:**
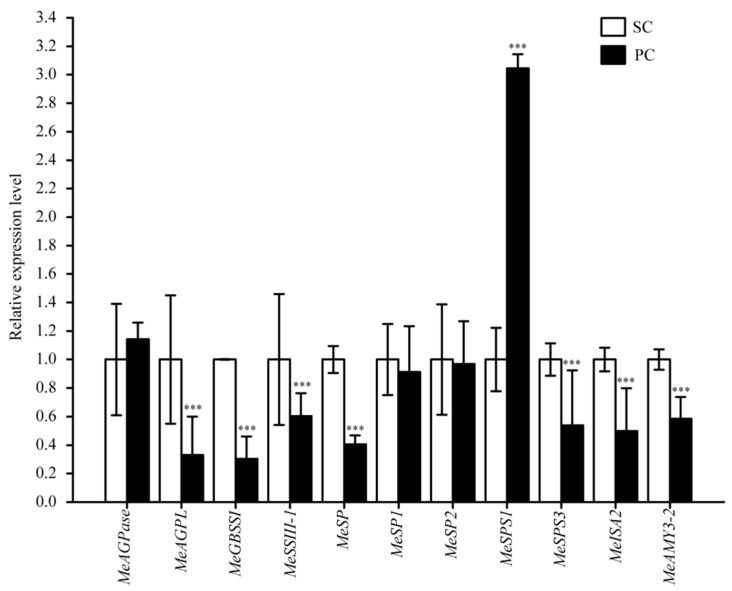
qRT-PCR results for starch biosynthetic and metabolic genes in PC compared with SC. The relative expression levels are normalized to *MeActin*. The x-axis represents the name of starch biosynthetic and metabolic genes. The y-axis represents the relative expression levels. The data represent the means of three biological replicates. All data are the means ± SE of three independent experiments. Significant differences are indicated by asterisk (*** *p* < 0.001). SC: *M. esculenta* Crantz cv. sugar cassava GPMS0991L; PC: *M. esculenta* Crantz cv. pink cassava BRA117315. *MeAGPase*, glucose-1-phosphate adenylyltransferase large subunit 1, *MeAGPL*, glucose-1-phosphate adenylyltransferase large subunit 3, *MeGBSSI*, granule-bound starch synthase, *MeSSIII-1*, starch synthase III-1, *MeSP*, glycogen phosphorylase, *MeSP1*, α-1,4 glucan phosphorylase 1, *MeSP2*, α-1,4 glucan phosphorylase 2, *MeSPS1*, sucrose-phosphate synthase 1, *MeSPS3*, sucrose-phosphate synthase 3, *MeISA2,* isoamylase 2, *MeAMY3-2*, alpha-amylase 3-2.

**Table 1 ijms-24-07236-t001:** Measurement of total starch, sucrose, glucose, and fructose of storage roots in different cassava germplasms.

Cassava Germplasms	The Content of Total Starch (%)	The Content of Sucrose (mg/kg)	The Content of Glucose (mg/kg)	The Content of Fructose (mg/kg)
SC	11.03 ± 0.05 b	12,850.65 ± 452.95 b	8725.70 ± 157.66 a	9874.10 ± 271.01 a
PC	29.04 ± 0.18 a	16,049.00 ± 115.97 a	1707.52 ± 54.79 b	1756.78 ± 36.99 b

Different letters (a and b) in a row indicate significant differences determined by the Duncan test (*p* < 0.05, *n* = 3), data are means ± standard errors. SC: *M. esculenta* Crantz cv. sugar cassava GPMS0991L; PC: *M. esculenta* Crantz cv. pink cassava BRA117315.

## Data Availability

All transcriptomic data are available at NCBI with the accession number PRJNA841388.

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
