# Peer review of "Integrated Metabolomic and Transcriptomic Analyses Reveals Sugar Transport and Starch Accumulation in Two Specific Germplasms of Manihot esculenta Crantz"

_ijms, 2023, doi:10.3390/ijms24087236_

Round 1
Reviewer 1 Report
The authors found that sugar cassava tubers contain higher levels of glucose and fructose and lower levels of starch and sucrose compared to pink cassava tubers. They claimed that the difference in composition was caused by different MeSweet gene expression in the two germplasms.
While this argument seems to be valid, I believe the result to support it is inadequate.
The main problems are as follows:
1. The carbohydrate in the tuber is transported the form of sucrose from the aboveground part through the phloem and stored as starch. In table I, suc + glu + fuc is around 30g/Kg in SC and 20g/Kg in PC. Converting the % starch amount to Kg is 110g starch/Kg in SC and 290g starch/Kg. This means that a much larger amount of carbohydrate moves to the tuber in PC. This explanation is supported by the much higher expression of sweet genes on PCs (Fig 5).
In this discussion, instead of focusing on the amount of starch, the author is focusing on the starch precursors GLU, FRU, and SUC, which are present in much smaller amounts. They are totally misunderstand the source-sink sugar partitioning.
2. Because SC has a high level of soluble sugar and a low level of storage insoluble sugar, it is possible that starch synthesis is slower than in PC. Many enzymes are involved in this pathway, as are the genes examined in Figure 7. However, there is insufficient information to draw any conclusions. AGPase is encoded by six genes in Arabidopsis and rice (2 small, 4 large subunit). There is no explanation for how many AGPase genes are present in cassava, whether they looked at tuber-specific expressed AGPase, or whether it is a large or small subunit. Is AGPL AGPase large subunit or a different gene? Other plants have numerous starch synthase genes. Is the tuber-specific gene MeSSIII-1? What exactly is SP? What exactly is SPS? They should thoroughly reconstruct, explain and discuss Fig 7.
3. Please show the levels of ADP-glucose, glucose-1-phosphate, and glucose-6-phosphate from the metabolic results, which are precursors to starch. This could be a hint as to which enzyme during starch synthesis is causing the low starch levels in the SC.
4. In the abstract, the “sugar cassava GPMS0991L and pink cassava BRA117315 were rich in sucrose and fructose, starch and glucose, respectively.” Glu and fuc are high in SC and starch and suc are high in PC in Table1.
5. What are a and b in Table 1?
6. The resolution of fig 2A and 2C are low.
Author Response
Please see the attachment, thank you very much.

Reviewer 2 Report
Since cassava from the tropical areas of the America has become a cash crop in many tropical and subtropical regions, it is important to cultivate and characterize the elite characteristics. The manuscript compared the difference in sugar transport and starch accumulation of the cassava storage roots in two specific germplasms. I found the work interesting. I only have a few concerns that need to be addressed before publication. In addition, I am concerned that the way how the authors interpreted their OIMCs data were unclear.
Figure 1. The two germplasms show significant difference in accumulation of carbohydrates, for phenotype analysis only the one transverse section image was shown. The longitudinal section should be presented.
Figure 2. It’s unclear how the volcano plot was produced, as shown in Figure 2A it seems there were far more than 299 colored dots.
Figure 3. How many genes were detected by Illumina sequencing? Which reference genome was used for annotation?
Figure 4. Describe in detail how correlation analysis of the metabolome and transcriptome was performed.
Figure 5 and 7. How to ensure the expression levels of Actin gene were same in two different germplasms if they were used as internal control. The calculated difference may originate from differential expression of Actin gene. At least one more internal control should be used.
Figure 6. Primers for MeSWEET genes amplification were missing.
Author Response

(The authors gave the same response as above.)

Reviewer 3 Report
I reviewed the article " Integrated metabolomic and transcriptomic analyses reveals sugar transport and starch accumulation in two specific germplasms of Manihot esculenta Crantz " and I found it interesting and scientifically valuable but poorly prepared. Research conducted by the Authors is undoubtedly interesting to read. In general, conceptualization was done accordingly to proper manners in the given field of science. Still the presentation of the results is on a medium level. In spite of that, it may be stated that the topic is an innovative one, which is essential in my opinion. Moreover, the background was nicely introduced in pinpointing crucial problems that need to be solved. Methods selected by the authors have been reasonably chosen according to the actual state of knowledge but its description are not very clear for me. As for the result section, it is worth mentioning that obtained results are indeed scientifically valuable. Still, the descriptive presentation of obtained outcomes could be done better. Altogether novelty and significance of content as well as scientific soundness of the manuscript are on a sufficient level. Unfortunately, the opportunity to highlight future key-needed research were not fulfilled at all. There are some minor corrections that have to be implemented before recommending the manuscript to be published. Overall, I recommend this manuscript to be published after minor revision.
Please see the detailed comments below:
I Introduction
The research background was presented clearly and transparently. It is clear which topics had to be broaden in order to enrich scientific knowledge. Given reasoning is easy to understand which makes the perception of the introduction very positive. Still, there is no clear indication of the research aims. Therefore, it is hard to follow the reason why and what the authors did to broaden the topic. Please do correct that.
II Results
I have mixed feelings after reading the results section. It has great “scientific soundness” I’d say but also it’s very difficult to comprehend. The reasons I see are as follow:
a)Each subsection is starting from general statements which are not the outcomes of the performed research. Hence should not be stated in the result section. In details - [66] – should be put in the introduction section; [75] – that should be in M&M section; [110-111] – that also should be in M&M section, [120-121]- these line should be in discussion section; [143-144] – should be placed in the introduction; [144-146] -should also be placed in introduction or in M&M section after restructuring.
b) All of the figures were placed out of the section. I find it very strange. In my opinion key figures should be placed within the text. Rest may stay as the supplementary materials at the end of the paper. Please do follow the recommendation of IJMS here.
All above makes the result section hard to read and comprehend. Please do correct that.
III Discussion
Discussion section was written properly. It has to be said that the presented statements follow the reasoning from previous section. It is also understandable why and what authors wanted to explain here. The chance to enrich the scientific knowledge with critical discussion using properly chosen references was used properly.
IV Materials and Methods:
The study methods were selected appropriately to the given research goal. The methodology itself will allow to recreate the conditions of the conducted experiment which I find crucial. This section was written nicely. It is easy to read and comprehend. Still I have some doubts about Statistical analysis.
[320-322] In here it is stated that all the statistical analysis were done with the Excell software but I can clearly see that some of the figures were prepared with the R-software. This has to be corrected. Please also cite the packages that were used for the analyses and visualisation. Moreover, Authors has mentioned only Kruskall-Wills test and students’ T test but it’s clear that far more data analysis were done. For instance Figure 4 was made based on Pearson correlation. That brings me to another doubts. Kruskall-Wills are generally used as a non parametric test were the data distribution does not fit the normality requirement, which is also essential for Pearson correlation to be conducted. Please do rewrite this subsection in the most detailed way ensuring clear perception of the analysis you conducted.
V Conclusions
The conclusions section was written poorly. I strongly recommend you to rewrite it pinpointing only the key findings of your study. Please do ensure that they are supported by the discussion section, which is crucial. Moreover, the opportunity to inform about the future research demands was not fulfilled at all. This should be corrected.
Author Response

(The authors gave the same response as above.)

Round 2
Reviewer 1 Report
- Please write the full name of the genes in the legend of Fig7.
- please explain what 'a' and 'b' mean in the legend of Table 1.
Author Response
The Answers for Reviewer 1
Question 1 from Reviewer 1: Please write the full name of the genes in the legend of Fig7.
Answer: Thanks for your comments. We added the full name of all genes in the legend of Figure 7. The revised parts were marked with blue color. Thanks again.
Question 2 from Reviewer 1: please explain what 'a' and 'b' mean in the legend of Table 1.
Answer: Thanks for your comments. We explain the meaning of 'a' and 'b' in the legend of Table 1. Thanks very much.
Other notes: We have recheck the writing and references, and revised in the revised version. Thanks again.

Reviewer 2 Report
Comments and suggestions:
1. Put images of the longitudinal section in Figure 1, although the difference is less than the cross section.
2. It is still unclear how modifications were made for volcano plot in Figure 2A by saying “We modified the Figure 2A, please see the new Figure 2 in the revised version.”
3. Same problem in Figure 3A as you have made in Figure 2A of previous version; it seems there were more than 9,077 genes (green and red dots) over the threshold you defined.
Author Response
Question 1 from Reviewer 2: The longitudinal section should be presented. Put images of the longitudinal section in Figure 1, although the difference is less than the cross section.
Answer: We accepted your suggestion and added the longitudinal section of SC and PC in Figure 1. Please see the new Figure 1 in revised version.
Question 2 from Reviewer 2: It is still unclear how modifications were made for volcano plot in Figure 2A by saying “We modified the Figure 2A, please see the new Figure 2 in the revised version.”
Answer: Thanks for your comments. A total of 2,705 metabolites (marked in Line 84) were used for Figure 2A, using the identification criterion of p ≤ 0.05 and VIP value ≥ 1, 299 metabolites in SC and PC (124 up-regulated and 175 down-regulated) were shown in the volcano map (with red and blue dots).
Question 3 from Reviewer 2: Same problem in Figure 3A as you have made in Figure 2A of previous version; it seems there were more than 9,077 genes (green and red dots) over the threshold you defined.
Answer: Thank you very much for your comments. For Figure 3A, a total of 23,268 genes (marked in Line 108) were detected by transcriptome sequencing. Through screening and filtering (p ≤ 0.05 and FC ≥ 2), 9,077 significantly differential genes (with green and red dots) were shown in the volcano map. We rechecked the transcriptome data, the volcano map was correct. Thanks for your reminder.
Other notes: We have recheck the writing and references, and revised in the revised version. Thanks again.